# Learning Disentangled Representations of Video with Missing Data

**Armand Comas Massagué**[1]
comasmassague.a@northeastern.edu

**Chi Zhang**[2]
zhang.chi13@northeastern.edu

**Zlatan Feric**[2]
feric.z@northeastern.edu

**Octavia Camps**[1]
camps@coe.neu.edu

**Rose Yu**[2,3]
roseyu@ucsd.edu [*]

## Abstract

Missing data poses significant challenges while learning representations of video sequences. We present Disentangled Imputed Video autoEncoder (DIVE), a deep generative model that imputes and predicts future video frames in the presence of missing data. Specifically, DIVE introduces a missingness latent variable, disentangles the hidden video representations into static and dynamic appearance, pose, and missingness factors for each object. DIVE imputes each object's trajectory where the data is missing. On a moving MNIST dataset with various missing scenarios, DIVE outperforms the state of the art baselines by a substantial margin. We also present comparisons on a real-world MOTSChallenge pedestrian dataset, which demonstrates the practical value of our method in a more realistic setting. Our code and data can be found at https://github.com/Rose-STL-Lab/DIVE.

## 1 Introduction

Videos contain rich structured information about our physical world. Learning representations from video enables intelligent machines to reason about the surroundings and it is essential to a range of tasks in machine learning and computer vision, including activity recognition [1], video prediction [2] and spatiotemporal reasoning [3]. One of the fundamental challenges in video representation learning is the high-dimensional, dynamic, multi-modal distribution of pixels. Recent research in deep generative models [4, 5, 6, 7] tackles the challenge by exploiting inductive biases of videos and projecting the high-dimensional data into substantially lower dimensional space. These methods search for *disentangled* representations by decomposing the latent representation of video frames into semantically meaningful factors [8].

Unfortunately, existing methods cannot reason about the objects when they are missing in videos. In contrast, a five month-old child can understand that objects continue to exist even when they are unseen, a phenomena known as "object permanence" [9]. Towards making intelligent machines, we study learning disentangled representations of videos with missing data. We consider a variety of missing scenarios that might occur in natural videos: objects can be partially occluded; objects can disappear in a scene and reappear; objects can also become missing while changing their size, shape, color and brightness. The ability to disentangle these factors and learn appropriate representations is an important step toward spatiotemporal decision making in complex environments.

In this work, we build on the deep generative model of DDPAE [5] which integrates structured graphical models into deep neural networks. Our model, which we call Disentangled-Imputed-Video-autoEncoder (DIVE), (**i**) learns representations that factorize into appearance, pose and missingness

---

[*][1]College of Electrical and Computer Engineering, [2] Khoury College of Computer Sciences, Northeastern University, MA, USA, [3]Computer Science & Engineering, University of California San Diego, CA, USA.

latent variables; (**ii**) imputes missing data by sampling from the learned latent variables; and (**iii**) performs unsupervised stochastic video prediction using the imputed hidden representation. Besides imputation, another salient feature of our model is (**iv**) its ability to robustly generate objects even when their appearances are changing by modeling the static and dynamic appearances separately. This makes our technique more applicable to real-world problems.

We demonstrate the effectiveness of our method on a moving MNIST dataset with a variety of missing data scenarios including partial occlusions, out of scene, and missing frames with varying appearances. We further evaluate on the Multi-Object Tracking and Segmentation (MOTSChallenge) object tracking and segmentation challenge dataset. We show that DIVE is able to accurately infer missing data, perform video imputation and reconstruct input frames and generate future predictions. Compared with baselines, our approach is robust to missing data and achieves significant improvements in video prediction performances.

## 2 Related Work

**Disentangled Representation.**    Unsupervised learning of disentangled representation for sequences generally falls into three categories: VAE-based [10, 6, 5, 7, 11, 12], GAN-like models [13, 14, 4, 15] and Sum-Product networks [11, 16]. For video data, a common practice is to encode a video frame into latent variables and disentangle the latent representation into *content* and *dynamics* factors. For example, [5] assumes the content (objects, background) of a video is fixed across frames, while the position of the content can change over time. In most cases, models can only handle complete video sequences without missing data. One exception is SQAIR [6], an generalization of AIR [17], which makes use of a latent variable to explicitly encode the *presence* of the respective object. SQAIR is further extended to an accelerated training scheme [16] or to better encode relational inductive biases [11, 12]. However, SQAIR and its extensions have no mechanism to recall an object. This leads to discovering an object as new when it reappears in the scene.

**Video Prediction.**    Conditioning on the past frames, video prediction models are trained to reconstruct the input sequence and predict future frames. Many video prediction methods use dynamical modeling [18] or deep neural networks to learn a deterministic transformation from input to output, including LSTM [19], Convolutional LSTM [20] and PredRNN [21]. These methods often suffer from blurry predictions and cannot properly model the inherently uncertain future [22]. In contrast to deterministic prediction, we prefer stochastic video prediction [2, 23, 22, 24, 14, 25], which is more suitable for capturing the stochastic dynamics of the environment. For instance, [22] proposes an auto-regressive model to generate pixels sequentially. [14] generalizes VAE to video data with a learned prior. [26] develops a normalizing flow video prediction model. [25] proposes a Bayesian Predictive Network to learn the prior distribution from noisy videos but without disentangled representations. Our main goal is to learn disentangled latent representations from video that are both interpretable and robust to missing data.

**Missing Value Imputation.**    Missing value imputation is the process of replacing the missing data in a sequence by an estimate of its true missing value. It is a central challenge of sequence modeling. Statistical methods often impose strong assumptions on the missing patterns. For example, mean/median averaging [27] and MICE [28], can only handle data missing at random. Latent variables models with the EM algorithm [29] can impute data missing not-at-random but are restricted to certain parametric models. Deep generative models offer a flexible framework of missing data imputation. For instance, [30, 31, 32] develop variants of recurrent neural networks to impute time series. [33, 34, 35] propose GAN-like models to learn missing patterns in multivariate time series. Unfortunately, to the best of our knowledge, all recent developments in generative modeling for missing value imputation have focused on low-dimensional time series, which are not directly applicable to high-dimensional video with complex scene dynamics.

## 3 Disentangled-Imputed-Video-autoEncoder (DIVE)

Videos often capture multiple objects moving with complex dynamics. For this work, we assume that each video has a maximum number of $N$ objects, we observe a video sequence up to $K$ time steps and aim to predict $T - K + 1$ time steps ahead. The key component of DIVE is based on

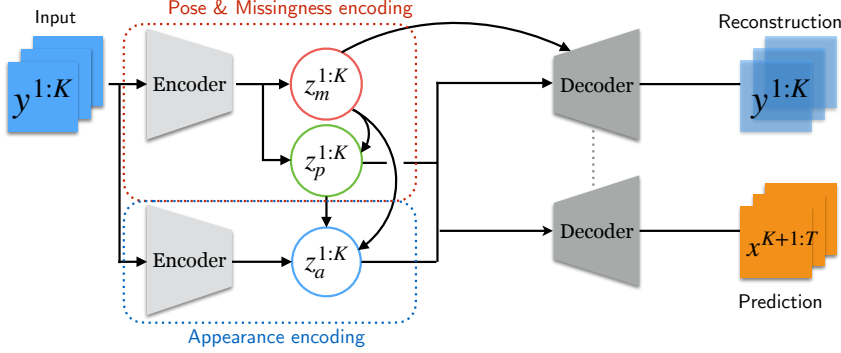

Figure 1: Overall architecture of DIVE, which takes the input video with missing data, infers the missingness (red), pose (green) and appearance (blue) latent variables. Two separate decoders reconstruct and predict the future sequences. The model is trained following the VAE framework.

the decomposition and disentangling of the objects representations within a VAE framework, with similar recursive modules as in [5]. Specifically, we decompose the objects in a video and assign three sets of latent variables to each object: appearance, pose and missingness, representing distinct attributes. During inference, DIVE encodes the input video into latent representations, performs sequence imputation in the latent space and updates the hidden representations. The generation model then samples from the latent variables to reconstruct and generate future predictions. Figure 1 depicts the overall pipeline of our model.

Denote a video sequence with missing data as $(\mathbf{y}^1, \cdots, \mathbf{y}^t)$ where each $\mathbf{y}^t \in \mathbb{R}^d$ is a frame. We assume an object in a video consists of *appearance*, *pose* (position and scale), and *missingness*. For each object $i$ in frame $t$, we aim to learn the latent representation $\mathbf{z}_i^t$ and disentangle it into three latent variables:

$$\mathbf{z}_i^t = [\mathbf{z}_{i,a}^t, \mathbf{z}_{i,p}^t, \mathbf{z}_{i,m}^t], \quad \mathbf{z}_{i,a}^t \in \mathbb{R}^h, \mathbf{z}_{i,p}^t \in \mathbb{R}^3, \mathbf{z}_{i,m}^t \in \mathbb{Z} \tag{1}$$

where $\mathbf{z}_{i,a}^t$ is the *appearance* vector with dimension $h$, $\mathbf{z}_{i,p}^t$ is the *pose* vector with $x$, $y$ coordinates and scale and $\mathbf{z}_{i,m}^t$ is the binary *missingness* label. $\mathbf{z}_{i,m}^t = 1$ if the object is occluded or missing.

## 3.1 Imputation Model

The imputation model leverages the missingness variable $\mathbf{z}_{i,m}^t$ to update the hidden states. When there is no missing data, the encoded hidden state, given the input frame, is $\mathbf{h}_{i,y}^t = f_{\text{enc}}(\mathbf{h}_{i,y}^{t-1}, \mathbf{h}_{i,y}^{t+1}, [\mathbf{y}^t, \mathbf{h}_{i-1,y}^t])$, where we enforce separate representations for each object. We implement the encoding function $f_{\text{enc}}$ with a bidirectional LSTM to propagate the hidden state over time. However, in the presence of missing data, such hidden state is unreliable and needs imputation. Denote the imputed hidden state as $\hat{\mathbf{h}}_{i,y}^t$ which will be discussed shortly. We update a latent space vector $\mathbf{u}_i^t$ to select the corresponding hidden state, given the sampled missingness variable:

$$\mathbf{u}_i^t = \begin{cases} \hat{\mathbf{h}}_{i,y}^t & \mathbf{z}_{i,m}^t = 1 \\ \gamma \mathbf{h}_{i,y}^t + (1 - \gamma) \hat{\mathbf{h}}_{i,y}^t & \mathbf{z}_{i,m}^t = 0 \end{cases}, \quad \gamma \sim \text{Bernoulli}(p) \tag{2}$$

Note that we apply a mixture of input hidden state $\mathbf{h}_{i,y}^t$ and imputed hidden state $\hat{\mathbf{h}}_{i,y}^t$ with probability $p$. In our experiments, we found this mixed strategy to be helpful in mitigating covariate shift [36]. It forces the model to learn the correct imputation with self-supervision, which is reminiscent of the scheduled sampling [37] technique for sequence prediction.

The pose hidden states $\mathbf{h}_{i,p}^t$ are obtained by propagating the updated latent representation through an LSTM network $\mathbf{h}_{i,p}^t = \text{LSTM}(\mathbf{h}_{i,p}^{t-1}, \mathbf{u}_i^t)$. For prediction we use an LSTM network, with only $h_{i,p}^{t-1}$ as input in time $t$. We obtain the imputed hidden state by means of auto-regression. This is based on the assumption that a video sequence is locally stationary and the most recent history is predictive of the

future. Given the updated latent representation at time $t$, the imputed hidden state at the next time step is:

$$\hat{\mathbf{h}}_{i,y}^t = \text{FC}(\mathbf{h}_{i,p}^{t-1}) \tag{3}$$

where $\text{FC}(\cdot)$ is a fully connected layer. This approach is similar in spirit to the time series imputation method in [32]. However, instead of imputing in the observation space, we perform imputation in the space of latent representations.

## 3.2 Inference Model

**Missingness Inference.** For the missingness variable $\mathbf{z}_{i,m}^t$, we also leverage the input encoding. We use a heaviside step function to make it binary:

$$\mathbf{z}_{i,m}^t = H(x), \quad x \sim \mathcal{N}(\mu_m, \sigma_m^2), \quad [\mu_m, \sigma_m^2] = \text{FC}(\mathbf{h}_{i,y}^t), \quad H(x) = \begin{cases} 1 & x \geq 0 \\ 0 & x < 0 \end{cases} \tag{4}$$

where $\sigma$ is the standard deviation of the noise, which is obtained from the hidden representation.

**Pose Inference.** The pose variable (position and scale) encodes the spatiotemporal dynamics of the video. We follow the variational inference technique for state-space representation of sequences [38]. That is, instead of directly inferring $\mathbf{z}_{i,p}^{1:K}$ for $K$ input frames, we use a stochastic variable $\beta_i^t$ to reparameterize the state transition probability:

$$q(\mathbf{z}_{i,p}^{1:T}|\mathbf{y}^{1:K}) = \prod_{t=1}^{K} q(\mathbf{z}_{i,p}^t|\mathbf{z}_{i,p}^{1:t-1}), \quad \mathbf{z}_{i,p}^t = f_{\text{tran}}(\mathbf{z}_{i,p}^{t-1}, \beta_i^t), \quad \beta_i^t \sim \mathcal{N}(\mu_p, \sigma_p^2) \tag{5}$$

where the state transition $f_{\text{tran}}$ is a deterministic mapping from the previous state to the next time step. The stochastic transition variable $\beta_i^t$ is sampled from a Gaussian distribution parameterized by a mean $\mu_p$ and variance $\sigma_p^2$ with $[\mu_p, \sigma_p^2] = \text{FC}(\mathbf{h}_{i,p}^t)$.

**Dynamic Appearance.** Another novel feature of our approach is its ability to robustly generate objects even when their appearances are changing across frames. $\mathbf{z}_{i,a}^t$ is the time-varying appearance. In particular, we decompose the appearance latent variable into a static component $\mathbf{a}_{i,s}$ and a dynamic component $\mathbf{a}_{i,d}$ which we model separately. The static component captures the inherent semantics of the object while the dynamic component models the nuanced variations in shape.

For the static component, we follow the procedure in [5] to perform inverse affine spatial transformation $\mathcal{T}^{-1}(\cdot; \cdot)$, given the pose of the object to center in the frame and rectify the images with a selected crop size. Future prediction is done in an autoregressive fashion:

$$\mathbf{a}_{i,s} = \text{FC}(\mathbf{h}_{i,a}^K), \quad \mathbf{h}_{i,a}^{t+1} = \begin{cases} \text{LSTM}_1(\mathbf{h}_{i,a}^t, \mathcal{T}^{-1}(\mathbf{y}^t; \mathbf{z}_{i,p}^t)) & t < K \\ \text{LSTM}_2(\mathbf{h}_{i,a}^t) & K \leq t < T \end{cases} \tag{6}$$

Here the appearance hidden state $\mathbf{h}_{i,a}^t$ is propagated through an LSTM, whose last output is used to infer the static appearance. Similar to poses, we use a state-space representation for the dynamic component, but directly model the difference in appearances, which helps stabilizing training:

$$\mathbf{a}_{i,d}^1 = \text{FC}([\mathbf{a}_{i,s}, \mathcal{T}^{-1}(\mathbf{y}^1; \mathbf{z}_{i,p}^1)]), \quad \mathbf{a}_{i,d}^{t+1} = \mathbf{a}_{i,d}^t + \delta_{i,d}^t, \quad \delta_{i,d}^t = \text{FC}([\mathbf{h}_{i,a}^t, \mathbf{a}_{i,s}]) \tag{7}$$

The final appearance variable is sampled from a Gaussian distribution parametrized by the concatenation of static and dynamic components, which are randomly mixed with a probability $p$:

$$q(\mathbf{z}_{i,a}|\mathbf{y}^{1:K}) = \prod_t \mathcal{N}(\mu_a, \sigma_a^2), \quad [\mu_a, \sigma_a^2] = \text{FC}([\mathbf{a}_{i,s}, \gamma \mathbf{a}_{i,d}^t]), \quad \gamma \sim \text{Bernoulli}(p) \tag{8}$$

The mixing strategy helps to mitigate covariate shift and enforces the static component to learn the inherent semantics of the objects across frames.

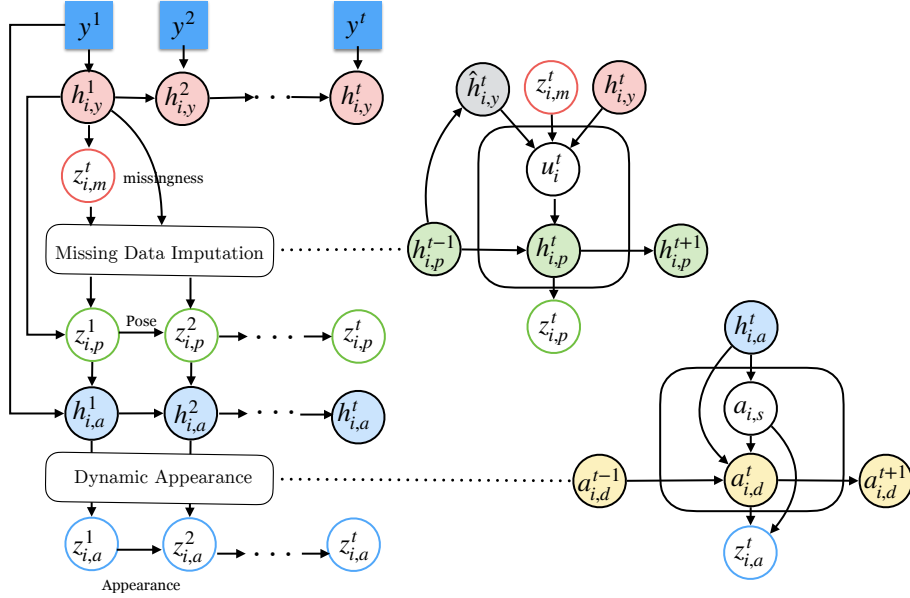

Figure 2: A graphical representation of DIVE. From top to bottom: inference of the missingness variable $\mathbf{z}_{i,m}^t$, missing data imputations model, inference of the pose vectors $\mathbf{z}_{i,p}^t$ and appearance variable $\mathbf{z}_{i,a}^t$ using dynamic appearance inference.

### 3.3 Generative Model and Learning

Given a video with missing data $(\mathbf{y}^1, \cdots, \mathbf{y}^t)$, denote the underlying complete video as $(\mathbf{x}^1, \cdots \mathbf{x}^t)$. Then, the generative distribution of the video sequence is given by:

$$p(\mathbf{y}^{1:K}, \mathbf{x}^{K+1:T} | \mathbf{z}^{1:T}) = \prod_{i=1}^{N} p(\mathbf{y}_i^{1:K} | \mathbf{z}_i^{1:K}) p(\mathbf{x}_i^{K+1:T} | \mathbf{z}_i^{K+1:T}) \quad (9)$$

In unsupervised learning of video representations, we simultaneously reconstruct the input video and predict future frames. Given the inferred latent variables, we generate $\mathbf{y}_i^t$ and predict $\mathbf{x}_i^t$ for each object sequentially. In particular, we first generate the rectified object in the center, given the appearance $\mathbf{z}_{i,a}^t$. The decoder is parameterized by a deconvolutional layer. After that, we apply an spatial transformer $\mathcal{T}$ to rescale and place the object according to the pose $\mathbf{z}_{i,p}^t$. For each object, the generative model is:

$$p(\mathbf{y}_i^t | \mathbf{z}_{i,a}^t) = \mathcal{T}(f_{\text{dec}}(\mathbf{z}_{i,a}^t); \mathbf{z}_{i,p}^t) \circ (1 - \mathbf{z}_{i,m}^t), \quad p(\mathbf{x}_i^t | \mathbf{z}_{i,a}^t) = \mathcal{T}(f_{\text{dec}}(\mathbf{z}_{i,a}^t), \mathbf{z}_{i,p}^t) \quad (10)$$

Future prediction is similar to reconstruction, except we assume the video is always complete. The generated frame $\mathbf{y}^t$ is the summation over $\mathbf{y}_i^t$ for all objects. Following the VAE framework, we train the model by maximizing the evidence lower bound (ELBO). Please see details in Appendix D .

## 4 Experiments

### 4.1 Experimental Setup

We evaluate our method on variations of moving MNIST and MOTSChallenge multi-object tracking datasets. The prediction task is to generate 10 future frames, given an input of 10 frames. The baselines include the established state-of-the-art video prediction methods based on disentangled representation learning: DRNET [4], DDPAE [5] and SQAIR [24].

**Evaluation Metrics.** We use common evaluation metrics for video quality on the visible pixels, which include pixel-level Binary Cross entropy (BCE) per frame, Mean Square Error (MSE) per

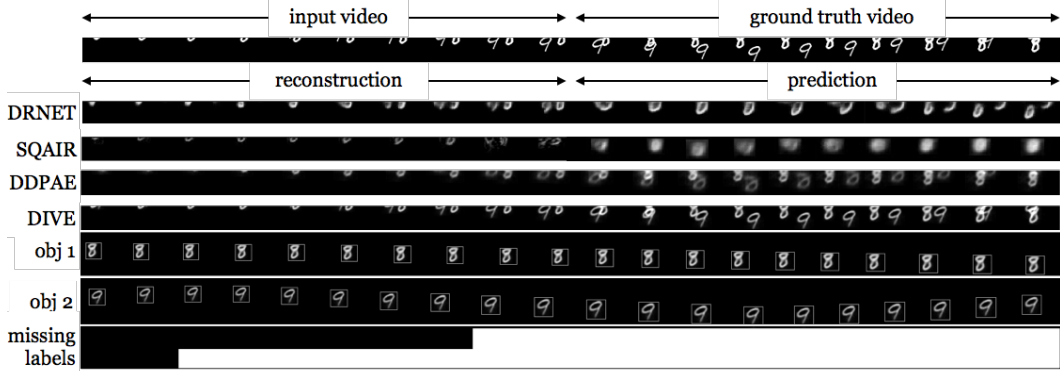

Figure 3: Partial missing qualitative results. *Obj 1* and *Obj 2* show DIVEs individual object generations and *missing labels* indicate whether each object is estimated completely missing in the scene. Note that objects are well decomposed, sharply generated and the labels properly predicted.

frame, Peak Signal to Noise Ratio (PSNR) and Structural Similarity (SSIM). Additionally, DIVE is a probabilistic model, hence we also report Negative Evidence Lower Bound (NELBO).

As our DIVE model simultaneously imputes missing data and generates improved predictions, we report reconstruction and prediction performances separately. For implementation details for the experiments, please see Appendix A.

## 4.2  Moving MNIST Experiments

**Data Description.**   Moving MNIST [19] is a synthetic dataset consisting of two digits with size $28 \times 28$ moving independently in a $64 \times 64$ frame. Each sequence is generated on-the-fly by sampling MNIST digits and synthesizing trajectories with fixed velocity with randomly sampled angle and initial position. We train the model for 300 epochs in scenarios 1 and 2, and 600 epochs in scenario 3. For each epoch we generate 10k sequences. The test set contains 1,024 fixed sequences. We simulate a variety of missing data scenarios including:

- *Partial Occlusion*: we occlude the upper 32 rows of the $64 \times 64$ pixel frame to simulate the effect of objects being partially occluded at the boundaries of the frame.
- *Out of Scene*: we randomly select an initial time step $t' = [3, 9]$ and remove the object from the frame in steps $t'$ and $t' + 1$ to simulate the out of scene phenomena for two consecutive steps.
- *Missing with Varying Appearance*: we apply an elastic transformation [39] to change the appearance of the objects individually. The transformation grid is chosen randomly for each sequence, and the parameter $\alpha$ of the deformation filter is set to $\alpha = 100$ and reduced linearly to 0 (no transformation) along the steps of the sequence. We remove each object for one time-step following the same logic as in scenario 2.

**Scenario 1: Partial occlusion.**    The top portion of Table 1 shows the quantitative performance comparison for all methods for the partial occlusion scenario. Our model outperforms all baseline models, except for the BCE in prediction. This is because DIVE generates sharper shapes which, in case of misalignment with the ground truth, have a larger effect on the pixel-level BCE. For reconstruction, our method often outperforms the baselines by a large margin, which highlights the significance of missing data imputation. Note that SQAIR performs well in reconstruction but fails in prediction. Prolonged full occlusions cause SQAIR to lose track of the object and re-identifying it as a new one when it reappears. Figure 3 shows a visualization of the predictions from DIVE and the baseline models. The bottom three rows show the decomposed representations from DIVE for each object and the missingness labels for objects in the corresponding order. We observe that DRNET and SQAIR fail to predict the objects position in the frame and appearance while DDPAE generates blurry predictions with the correct pose. These failure cases rarely occur for DIVE.

**Scenario 2: Out of Scene.**    The middle portion of Table 1 illustrates the quantitative performance of all methods for scenario 2. We observe that our method achieves significant improvement across all metrics. This implies that our imputation of missing data is accurate and can drastically improve the predictions. Figure 4 shows the prediction results of all methods evaluated for the out of scene case. We observe that DRNET and SQAIR fail to predict the future pose, and the quality of the

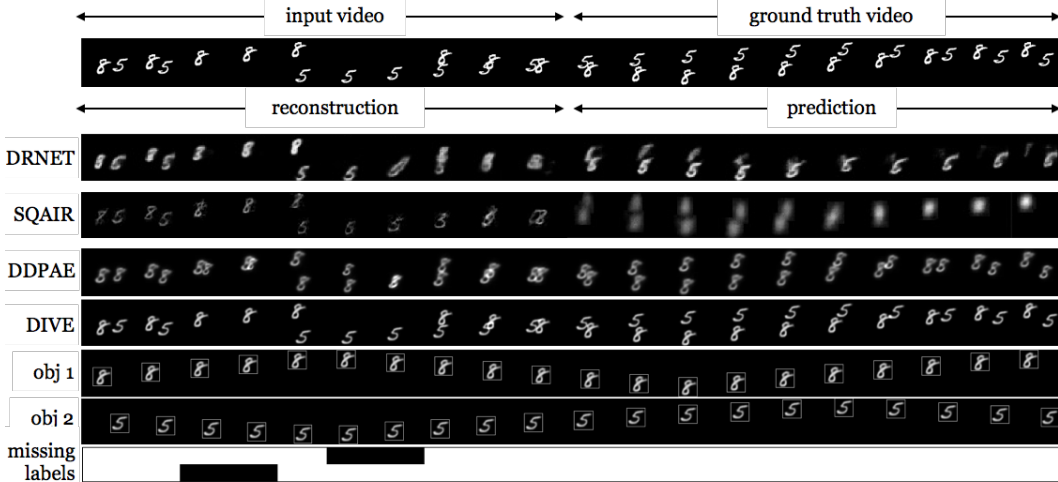

Figure 4: Qualitative results for out of scene missing scenario for two time steps.

generated object appearance is poor. The qualitative comparison with DDPAE reveals that the objects generated by our model have higher brightness and sharpness. As the baselines cannot infer the object missingness, they may misidentify the missing object as any other object that is present. This would lead to confusion for modeling the pose and appearance. The figure also reveals how DIVE is able to predict the missing labels and hallucinate the pose of the objects when missing, allowing for accurate predictions.

**Scenario 3: Missing with Varying Appearance.** Quantitative results for 1 time step complete missingness with varying appearance are shown in the bottom portion of Table 1. Our method again achieves the best performance for all metrics. The difference between our models and baselines is quite significant given the difficulty of the task. Besides the complete missing frame, the varying appearances of the objects introduce an additional layer of complexity which can misguide the inference. Despite these challenges, DIVE can learn the appearance variation and successfully recognize the correct object in most cases. Figure 5 visualizes the model predictions, a tough case where two seemingly different digits ("2" and "6") are progressively transformed into the same digit ("6"). SQAIR and DRNET have the ability to model varying appearance, but fail to generate

Table 1: Quantities comparison of all methods for three missing scenarios w.r.t. reconstruction and prediction. From top to bottom: partial occlusion, out of scene and complete missing with varying appearance. The improvements of our method DIVE are evident for all scenarios.

| Scenario 1 | BCE ↓ | | MSE ↓ | | PSNR ↑ | | SSIM ↑ | | NELBO ↓ |
|---|---|---|---|---|---|---|---|---|---|
| Model | Rec | Pred | Rec | Pred | Rec | Pred | Rec | Pred | |
| DRNET[4] | 482.07 | 852.59 | 72.21 | 96.36 | 7.99 | 6.89 | 0.76 | 0.72 | / |
| SQAIR[6] | 178.71 | 967.20 | 21.84 | 84.73 | 13.19 | 9.96 | **0.90** | 0.73 | -0.16 |
| DDPAE[5] | 182.66 | **417.00** | 39.09 | 67.41 | 17.56 | 15.49 | 0.77 | 0.72 | -0.09 |
| **DIVE** | **119.25** | 459.10 | **19.73** | **64.49** | **20.64** | **15.85** | **0.90** | **0.78** | **-0.18** |
| Scenario 2 | | | | | | | | | |
| DRNET | 392.33 | 1402.45 | 90.64 | 187.72 | 9.59 | 9.88 | 0.80 | 0.67 | / |
| SQAIR | 468.22 | 927.09 | 73.13 | 137.04 | 10.33 | 8.21 | 0.84 | 0.69 | -0.17 |
| DDPAE | 266.03 | 409.26 | 58.37 | 89.57 | 18.64 | 16.94 | 0.87 | 0.77 | -0.17 |
| **DIVE** | **165.42** | **321.29** | **27.03** | **64.17** | **22.15** | **18.56** | **0.93** | **0.83** | **-0.21** |
| Scenario 3 | | | | | | | | | |
| DRNET | 421.72 | 1304.53 | 90.46 | 176.28 | 9.91 | 7.33 | 0.75 | 0.70 | / |
| SQAIR | 560.51 | 1518.61 | 74.30 | 163.25 | 10.80 | 7.64 | 0.83 | 0.62 | -0.16 |
| DDPAE | 322.23 | 403.48 | 63.63 | 82.71 | 18.29 | 17.22 | 0.81 | **0.78** | -0.18 |
| **DIVE** | **272.74** | **374.59** | **42.81** | **74.87** | **20.08** | **17.61** | **0.87** | **0.78** | **-0.19** |

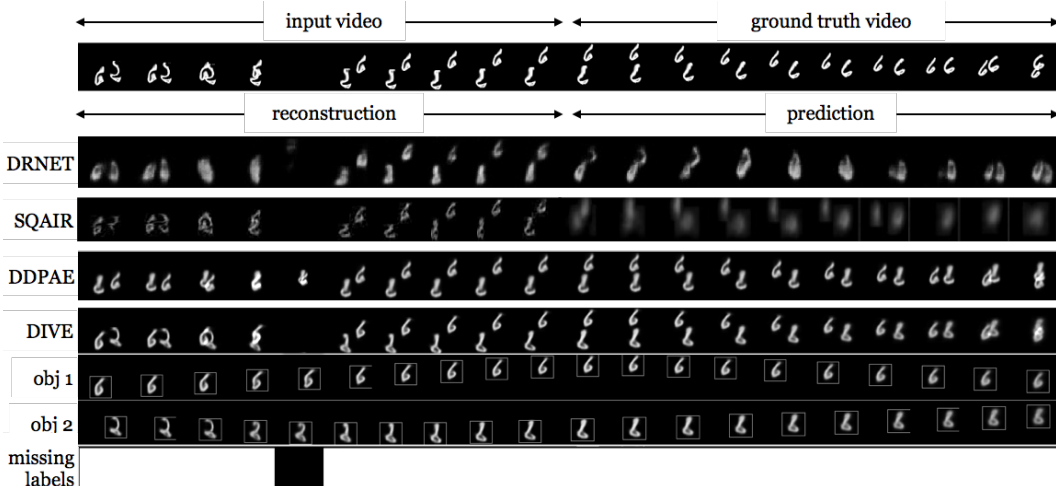

Figure 5: Qualitative results for one time step complete missing with varying appearance.

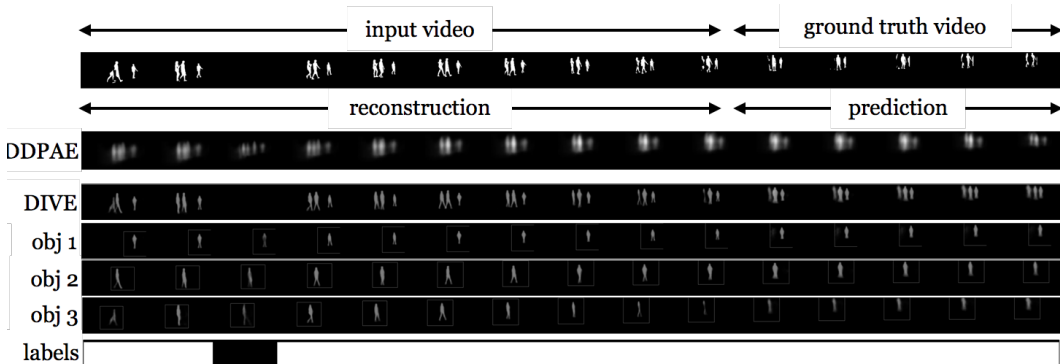

Figure 6: MOTS data set qualitative results. Note that our method successfully identifies the missing time step, decomposes the objects and keeps track of the missing pedestrians.

reasonable predictions due to similar reasons as before. DDPAE correctly predicts the pose after the missing step, but misidentifies the objects appearance before that. Also, DDPAE simply cannot model appearance variation. DIVE correctly estimates the pose and appearance variation of each object, while maintaining their identity throughout the sequence.

## 4.3 Pedestrian Experiments

The Multi-Object Tracking and Segmentation (MOTS) Challenge [40] dataset consists of real world video sequences of pedestrians and cars. We use 2 ground truth sequences in which pedestrians have been fully segmented and annotated [41]. The annotated sequences are further processed into shorter 20 frame sub-sequences, binarized and with at most 3 unique pedestrians. The smallest objects are scaled and the sequences are augmented by simulating constant camera motion and 1 time step complete camera occlusion, further details deferred to Appendix B.

Table 2 shows the quantitative metrics compared with the best performing baseline DDPAE. This dataset mimics the missing scenarios 1 (partial occlusion) and 3 (missing with varying appearance) because the appearance walking pedestrians is constantly changing. DIVE outperforms

Table 2: Quantitative comparison on MOTS pedestrian dataset for DDPAE and DIVE.

| Model | BCE ↓ | MSE ↓ | PSNR ↑ | SSIM ↑ | NELBO ↓ |
|-------|-------|-------|--------|--------|---------|
| DDPAE | 2495.08 | 560.37 | 22.22 | 0.90 | -0.24 |
| DIVE | **1355.89** | **328.96** | **24.82** | **0.96** | **-0.26** |

DDPAE across all evaluation metrics. Figure 6 shows the outputs from both models as well as the decomposed objects and missingness labels from DIVE. Our method can accurately recognize 3 objects (pedestrians), infer their missingness and estimate their varying appearance. DDPAE fails to

decompose them due to its rigid assumption of fixed appearances and the inherent complexity of the scenario. In Appendix C, we perform two ablation studies. One on the significance of dynamic appearance modeling, and the other on the importance of estimating missingness and performing imputation.

## 5    Conclusion and Discussion

We propose a novel deep generative model that can simultaneously perform object decomposition, latent space disentangling, missing data imputation, and video forecasting. The key novelty of our method includes missing data detection and imputation in the hidden representations, as well as a robust way of dealing with dynamic appearances. Extensive experiments on moving MNIST demonstrate that DIVE can impute missing data without supervision and generate videos of significantly higher quality. Future work will focus on improving our model so that it is able to handle the complexity and dynamics in real world videos with unknown object number and colored scenes.

## Broader Impact

Videos provide a window into the physics of the world we live in. They contain abundant visual information of what objects are, how they move, and what happens when cameras move against the scene. Being able to learn a representation that disentangles these factors is fundamental to AI that can understand and act in spatiotemporal environment. Despite the wealth of methods for video prediction, state-of-the-art approaches are sensitive to missing data, which are very common in real-world videos. Our proposed model significantly improves the robustness of video prediction methods against missing data, and thereby increasing the practical values of video prediction techniques and our trust in AI. Video surveillance systems can be potentially abused for discriminatory targeting, and we remained cognizant of the bias in our training data. To reduce the potential risk of this, we pre-processed the MOTSChallenge videos to greyscale.

## Acknowledgments and Disclosure of Funding

This work was supported in part by NSF under Grants IIS#1850349, IIS#1814631, ECCS#1808381 and CMMI#1638234, the U. S. Army Research Office under Grant W911NF-20-1-0334 and the Alert DHS Center of Excellence under Award Number 2013-ST-061-ED0001. The views and conclusions contained in this document are those of the authors and should not be interpreted as necessarily representing the official policies, either expressed or implied, of the U.S. Department of Homeland Security. We thank Dr. Adam Kosiorek for helpful discussions. Additional revenues related to this work: ONR # N68335-19-C-0310, Google Faculty Research Award, Adobe Data Science Research Awards, GPUs donated by NVIDIA, and computing allocation awarded by DOE.

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
