[Supplementary Material]

# A  Model Implementation Details

**SQAIR**    The SQAIR model is sensitive to hyper-parameters [6]. Different combinations of hyper-parameters are used to reproduce the best performance of the model. However, through our communication with the authors, SQAIR model is not designed for the missing data scenario. Thus, we were not able to reach similar level of performance reported in [6]. In order to obtain the best performance of SQAIR for our data set, we trained and evaluated the reconstruction model and prediction model separately as we found that SQAIR model is more stable with the reconstruction task and thus could be trained longer (300 epochs and more). However, for the prediction task, the main issue we encountered was that the gradients would vanish for some combinations of hyper-parameters and the model was not able to make predictions after certain number of training epochs (this number can fluctuate). In order to obtain the best performance of SQAIR, we kept the model training until it could generate predictions and select the checkpoint with the best performance. We used rmsprop to optimize the SQAIR model and the model uses important-weighted auto-encoder[42] with 5 particles as a general structure. For more implementation details, please refer to the github of SQAIR[6]. It is also important to note that the training time per 100 epochs for SQAIR is at least 5 times more than the training time of our DIVE model.

**DRNET**    The original version of DRNET model only uses the first four frames for training. In order to adapt DRNET for our prediction task, we changed the scene discriminator in DRNET to train on all frames in the sequences. This modification is more suitable for our missing data scenario. Because if there were missing data in the first four frames, the scene discriminator trained only on the first four frames would easily fail. However, after this modification, the probability for the scene discriminator to successfully recognize the scene also increases. Except for this modification, the rest of the model was kept exactly the same as the author's implementation for better reproduction of results. It is also important to note that the main network and the LSTM in DRNET were trained separately. The main network was trained first and then the LSTM was trained based on results of the main network. Therefore, if the main network failed to recognize the objects, the LSTM would also fail to learn the trajectories. We used the default Adam optimizer in DRNET to train the model. The scene discriminator was trained with BCE loss. The main network and LSTM were trained with MSE loss. For more implementation details, please refer to the github of DRNET[4].

**DDPAE**    We used the code provided by the authors. The hyperparameters that they use in the public version were kept unchanged. Also we followed the instructions in their github repository (`https://github.com/jthsieh/DDPAE-video-prediction`) for the Moving MNIST experiment. However, for some of the experiments, we have added additional features to produce better results. For the Pedestrian experiments, we aligned the hyperparameters that are semantically similar with DIVE implementation. Also, the pose size was constrained less than in the default setting. This way, the model can adapt to a highly varying dataset.

**DIVE**    The main variables have the following dimensions: $\mathbf{z}_{i,a}^t \in \mathbb{R}^{128}$, $\mathbf{a}_{i,s} \in \mathbb{R}^{256}$, $\mathbf{a}_{i,d}^t \in \mathbb{R}^{48}$, $\mathbf{z}_{i,p}^t \in \mathbb{R}^3$, $\mathbf{z}_{i,m}^t \in \mathbb{Z}^1$ and $\mathbf{h}_{i,y}^t \in \mathbb{R}^{64}$. The dimensions were chosen after a manual sweep of hyperparameters range. Particularly, the dimensionality of $\mathbf{a}_{i,d}^t$ was chosen from the range $[12, 24, 48, 64]$; $\mathbf{z}_{i,a}^t$ and $\mathbf{a}_{i,s}$ from $[64, 128, 256]$; and $\mathbf{h}_{i,y}^t$ from $[48, 64, 96]$. The learning rate was set to $10^{-3}$ and reduced to $4^{-3}$ at $1/3$ of the training iterations, and we used a batch size of $64$. The Bernoulli distribution for the imputation model has probability $p = 0.25$ in training and $p = 0$ in testing. For the appearance model, the Bernoulli distribution has $p = 0.7$, which was increased to $p = 0.85$ after 3,000 iterations during training. For testing, we set $p = 1$. In both cases, we experimented with different values of p ($\pm 0.1$) and did not find any significant difference. The number of objects $N$ is specified a priori, but only as an upper bound. For the MOTS dataset, we set $N = 3$ but the actual number of objects is often lower. For all the Moving MNIST experiments we set $N = 2$. Similarly to DDPAE, our model can learn to set the redundant components to be empty. Further details can be found in the provided codebase.

The missingness latent variable, $\mathbf{z}_{i,m}^t$, was implemented with a Heaviside step function in the pose encoding model, with a $-0.5$ bias in the logit. However, to allow the gradients to propagate, the Heavyside function $H(x)$ is not a variable node in our computational graph, hence we are not differentiating through it. We use *torch.where()* function in Pytorch to implement this condition operation. As shown in Figure 1 (top) and Equation 10, $\mathbf{z}_{i,m}$ is a masking indicator. In practice, we

multiply each decoded object by the logit before the Heavyside function instead of the binary label. Hence, $\mathbf{z}_{i,m}$ gets its gradients from the decoder.

To adapt to three missing scenarios, we made minor changes to our implementation. For missing scenario 1 (partial occlusion) and 2 (out of scene) of the MovingMNIST experiments, because the objects appearance remain static, we did not include the dynamic appearance model component. The appearance encoding is therefore adjusted accordingly. We followed Equation 6 to generate the static appearance, but we skipped the input frames $\mathbf{y}^t$ and hidden states $\mathbf{h}^t_{i,a}$ in LSTM$_1$ where we predicted missingness $\mathbf{z}^t_{i,m} = 1$. For partial occlusion training with Moving MNIST dataset, we used a scheduling mechanism to evaluate the loss only for the visible area of the frame. We applied the same procedure to all the baselines for a fair comparison. For the pedestrian dataset, similarly to DDPAE, we relaxed the pose size constraint to accommodate the highly dynamic pose size in real-world videos. With this implementation, we measure the training time. It takes 91 minutes to carry out 100 epochs, for which we process 1 million samples in batches of 64.

**Software**   We implemented this method using Ubuntu 18.04, Python 3.6, Pytorch 1.2.0, Cuda 10.0 and Pyro 0.2 as a framework for probabilistic programming.

**Hardware**   For each of our experiments we used 1 GPU RTX 2080 Ti (Blower Edition) with 128GB of memory.

## B   Datasets Details

**Moving MNIST with elastic deformation.**   In order to simulate slowly varying appearance in Scenario 3, we applied an elastic deformation to the objects in the scene. Given a uniform grid that represents the object pixel coordinates, we generated a distortion. We created a displacement random field with parameters $\alpha$ and $\sigma$. These parameters controlled the intensity of the deformation and the smoothing of the field, respectively. The displacement field was added to the uniform grid, and used to deform the coordinates of the given digit. This is described in [39]. The transformation was done independently to every digit. We set $\sigma = 4$ and $\alpha$ varied linearly from 100 to 0 along the sequence.

**MOTS Challenge pre-processing.**   The Multi-Object Tracking and Segmentation (MOTS) Challenge [40] dataset focuses on the task of multi-object tracking to multi-object tracking and segmentation. It provides dense pixel-level annotations for two existing tracking datasets. It comprises 65,213 pixel masks for 977 distinct objects (cars and pedestrians) in 10,870 video frames. For our task, we used 2 scenes with only pedestrians. Each one of these scenes was processed as follows: We kept the dense annotations as the shapes of the objects, and discarded all remaining content (such as the background). Given the large variance in the objects size, we resized the objects below the average size in the scene to the average and added a small random margin. Each scene was divided in sequences of 20 frames, reducing the sampling rate by a factor of 5 to increase displacements of objects. For each sequence, we selected all combinations of 3 objects to augment the data. We binarized the grey values of all sequences. Each sequence was padded randomly to fit a square and resized to $256 \times 256$ pixels. Finally, during training we added on-the-fly transformation to the clips. We subtracted all content for one random time step and sequentially affine-transformed the frames to simulate full camera occlusion and constant camera motion. This was also done when generating the fixed testing sequences. As a result, we used 4,416 sequences for training and 675 for testing, while making sure they belonged to different scenes.

## C   Ablation Study

### C.1   Appearance variability

In order to highlight the significance of dynamic appearance modeling, we performed an ablative study for DIVE, focusing on Scenario 3 in Section 4.2. In particular, we compared two cases: (1) **dynamic appearance.** This is our main configuration. Missingness was estimated with hard labels, binarized with a step function while encoding. The appearance was modeled as in Equation (8), where the Bernoulli probability is $p = 1$ in testing and therefore we explicitly modeled the dynamic appearance. (2) **static appearance.** In this case, we altered the original configuration by setting

Figure 7: Ablation study for static and dynamic appearance modeling for missing Scenario 3 of the Moving MNIST experiments. DDPAE results were also shown for comparison purposes.

$p = 0$ for the Bernoulli distribution in Equation 8. This allows only for static (constant) appearance generation.

For each case, we trained the model for 600 epochs and kept the model every 100 epochs for the range $[200, 600]$, with the same training and testing setup as previously reported for this scenario. We used the DDPAE results trained for 600 epochs as a baseline. We tested the models and report BCE and MSE per frame metrics, separately for input reconstruction and output prediction.

Figure 7 shows the quantitative results for both BCE (left) and MSE (right). We can see that for reconstruction, having dynamic appearance components significantly reduces the reconstruction error, specially for MSE. This is because Scenario 3 contains digits with large distortion in the input. Hence more flexible appearance modeling better adapts to the changing shapes.

However, predicting the sequence into the future inevitably introduces uncertainty, leading to blurry predictions. Static modeling captures the shared constant appearance, which have low intensity deformations. Therefore, it does not suffer large appearance variations, and generates sharper shapes in prediction. However, as the baseline DDPAE does not provide a mechanism for missing data imputation or varying appearance, both of our approaches outperform (DDPAE) by a large margin, even in the early stages of training.

We also conducted an ablative study on the missingness variable. In our implementation, we chose a heavy-sided function to obtain "hard" missingness labels. An alternative is a *Sigmoid* activation function to obtain "soft" label $\mathbf{z}_{i,m}^t \in (0, 1)$. We tested both and found that the model can always learn the labels correctly. The performance difference was not statistically significant.

## C.2 Missingness Variable

In our experiments, we consider DDPAE to be the closest architecture to ours without missingness variables. To further validate the significance of explicitly modeling missingness, we also test the exact same DIVE architecture with and without the missingness variable, for Scenario 2 of Moving MNIST experiments. Table 3 shows the quantitative results after 300 epochs of training:

Table 3: Quantitative results for Scenario 2 of Moving MNIST experiments. We perform experiments with and without missing data label and imputation. Missing data imputation for DIVE shows clear improvements in all metrics for both reconstruction and prediction.

| Mov. MNIST Scenario 2 | BCE ↓ | | MSE ↓ | | PSNR ↑ | | SSIM ↑ | |
|---|---|---|---|---|---|---|---|---|
| Model (trained 300 epochs) | Rec | Pred | Rec | Pred | Rec | Pred | Rec | Pred |
| **DIVE w/o missingness** | 236.35 | 356.82 | 49.07 | 76.52 | 19.40 | 17.66 | 0.86 | 0.82 |
| **DIVE w missingness** | **165.42** | **321.29** | **27.03** | **64.17** | **22.15** | **18.56** | **0.93** | **0.83** |

The results demonstrate that explicitly reasoning about missingness and performing imputation is the key, not only to the reconstruction but also to future predictions. This is partially attributed to a better learning of the underlying dynamics of the scene.

## D  Optimization Objective

The optimization objective is to maximize the evidence lower bound (ELBO), as in the common VAE framework:

$$
\begin{aligned}
\log p_\theta(\mathbf{y}^{1:K}, \mathbf{x}^{K+1:T}) \;\geq\; & \mathbb{E}_q \left[ \log p_\theta \left( \mathbf{y}^{1:K} | \mathbf{z}_{1:N}^{1:K} \right) - \mathrm{KL} \left( q_\phi \left( \mathbf{z}_{1:N}^{1:K} \right) || p \left( \mathbf{z}_{1:N}^{1:K} \right) \right) \right] \qquad (11) \\
& + \; \mathbb{E}_q \left[ \log p_\theta \left( \mathbf{x}^{K+1:T} | \mathbf{z}_{1:N}^{K+1:T} \right) - \mathrm{KL} \left( q_\phi \left( \mathbf{z}_{1:N}^{K+1:T} \right) || p \left( \mathbf{z}_{1:N}^{K+1:T} \right) \right) \right]
\end{aligned}
$$

Here, DIVE uses self-supervision for reconstructing the corrupted input $\mathbf{y}^{1:K}$ and predicting the complete output $\mathbf{x}^{K+1:T}$. We add a regularization term to minimize the KL-divergence between our latent space representation and a Gaussian prior, parametrized by its mean and variance. Note that $N$ is our prior on the number of objects in the scene.

## E  More examples and failure cases of DIVE

In this section, we provide more examples including failure cases from three missing scenarios experiments. For each of the examples, the first 10 frames are the inputs, followed by the 10 predicted frames. The top row is the ground truth and the second to the last row is the reconstructions/predictions from DIVE. We also show the decomposed objects and the learned missingness labels, respectively.

(a) A failure case where our DIVE model cannot reconstruct and predict digit "7", as it doesn't appear in the input.

(b) A success case where our model recovers the heavily corrupted digit.

Figure 8: More examples for missing Scenario 1: Partial occlusion experiment. The rows for each figure from top to bottom are (1) ground truth, (2) first object, (3) second object, (4) DIVE predictions, (5) predicted missing labels for each object. We use the same display format for all Moving MNIST examples below.

Figure 8 shows three examples for missing scenario 1 (partial occlusion). Figure 8(a) shows a failure case where DIVE cannot recognize and generate digit "7" as it only reappears at the very end. This is partially due to our imputation mechanism, which only uses the previous information not the future information. Figure 8(b) shows a success case where even though one of the digits is heavily corrupted in the input frames, DIVE could still reconstruct it in the results. In this case, digit "5" is totally missing in five input frames and is heavily corrupted or overlaps with the other digits in the rest of the input frames. Our DIVE model successfully reconstructs and predicts it in almost all frames. It is also important to note that the imputation of the missing digit five from second frame to seventh frame is smooth and accurate (see third row of the figure).

(a) Failure case with two digits overlapping in all input frames.

(b) Success case with two digits overlapping frequently.

Figure 9: Examples for Scenario 2: Out of scene for two time steps.

(a) A failure case where the model misrecognizes the digits.

(b) A success case on two similar digits.

Figure 10: Examples for Scenario 3: varying appearance experiment.

Figure 9 shows more examples from missing Scenario 2 (out of scene). Specifically, a failure case where DIVE cannot recognize both of the digits is shown in Figure 9(a). In this case, the digits entangle with each other almost in every frame and thus the model recognizes them as one object. We also show a success case where the two digits entangle with each other frequently in Figure 9(b). From these two cases, we can conclude that the model needs as least one frame where the two digits are separable to generate decent results.

Figure 10 displays more examples for an experiment on missing Scenario 3 (complete missing with varying appearance). Figure 10(a) shows a failure case where after the fifth frame, our DIVE model

mis-recognizes the two digits. The switching happens when in the fifth frame, digit "8" is missing from the scene and the digit "8" and "0" have similar appearances. After the switching, the model fails to recover the initial assignment of the objects. Although in this example our model generates decent results, we cannot overlook the potential issue. Especially when the trajectories of objects are very complex and heterogeneous, confusion in appearances could lead to inaccurate predictions of trajectories. Figure 10(b) shows a success case where the two digits are similar.

←——————————— reconstruction ——————————→  ←——— prediction ———→

(a) A failure case with a split object.

(b) A success in a difficult case of overlapping objects.

Figure 11: Qualitative examples for pedestrian (MOTS) dataset. The rows from top to bottom are: (1) ground truth, (2) first object, (3) second object, (4) third object, (5) DIVE prediction, (6) combined predicted missing labels for each object.

More examples from the MOTS data set are shown in Figure 11. Figure 11(a) shows a failure example where the object/pedestrian is partly present in the $4_{th}$ row, that should be empty. Given the low displacement of the objects, the model sometimes has problems to infer which entities are independent. This can create duplicated content when we decompose the frames. It can also happen, that two objects that are static or move at the same velocity are encoded as a single entity. The failure case also shows how the model can't predict the appearance of a new object that hasn't been identified in the input. This is not surprising, as a human wouldn't have been able to make such prediction. Figure 11(b) shows a success case where our model encodes each pedestrian properly and generates reasonable predictions. This case is especially difficult because, although there is no full frame missing, two of the objects overlap for several frames at the input sequence.