[Reviews · NeurIPS 2020]

Review 1

Summary and Contributions: This paper proposes a novel VAE-based stochastic video generation model that decomposes the latent space into object-specific latent variables that encode appearance, pose, and missingness (i.e. is the object present in the frame). The authors compare their approach with a variety of other methods on the moving MNIST dataset and a grayscale/binarized version on the MOTS dataset. They demonstrate their method accurately infers the binary missingness variable and effectively predicts future frame under different settings.

Strengths: This paper is well written and well motivated, for the most part. The model is a sensible extension of previous approaches and performs adequately on simple datasets.

Weaknesses: This paper would significantly benefit from greater experimental evaluation and analysis. There are some standard video prediction datasets missing from comparison, and there are many ablation studies / analysis of the different model components that I'd encourage the authors to consider. (More details on my recs below.)

Correctness: Yes.

Clarity: Easy to follow, well written.

Relation to Prior Work: Yes.

Reproducibility: Yes

Additional Feedback: - The authors refer to DrNet as a SOTA stochastic video generation method. However, DrNet is not a stochastic video prediction method, making it a somewhat inappropriate comparison for the stochastic dataset. - The authors say "we enforce separate representations for each object." -- could you specify how this is done? Is the number of objects specified for the model a priori? - Given that the key novelty in this method arises from the missingness variable, I would suggest exploring some dataset (natural or artificial) that test the limits / effectiveness of this new component better. For example, a dataset that occludes an object behind another object would strengthen the paper. - It would be good to see an ablation study where the exact same model architecture *without* the missingness variable is compared. More generally, this model has several different components and additional analysis of the strengths / importance of each of these components would strengthen the paper. - The ability of this model to track multiple objects is a strength, however this currently feels under explored. How many objects can the model handle, and at what point does it break down? It would also be good to evaluate the method on a well used video prediction dataset with multiple objects (but not necessarily occlusion) to see if it performs as well as (or better) that current SOTA approaches. For example, the BAIR pushing dataset, while having few moments of occlusion, does have multiple moving objects and is also a well used dataset and comparison on that could strengthen the results.


Review 2

Summary and Contributions: This paper presents a deep generative model of video sequences by decomposing the the latent representations of videos into factors while accounting for the ability to reason about objects that are missing in videos or occluded. The key idea of the DIVE model proposed herein is to: 1) factorize the representation (of each frame) into appearance, pose and missingness 2) impute data when missing 3) use the model for video prediction by modeling the static and dynamic objects separately An bidirectional LSTM is used to encode each frame into a representation. A separate representation is inferred for each of the different objects. Then, a univariate Normal distribution is used to decide whether or not the object is missing or present. A sample from this distribution is passed through a step function to obtain a binary representation of whether or not the object is missing. The hidden state of the object if it is missing is set based either on the current representation or the previous representation of the object based on a hyperparameter. This "missingness corrected" representation is passed through an LSTM to obtain the pose representation. A time-varying dynamic and static representation for the appearance are also inferred. Using these disentangled representations of each object in each frame, two different LSTMs are used to parameterize the reconstruction and future prediction prediction of frames. The decoding processes uses a spatial transformer and the inferred pose variable to rescale and place the object into the frame.

Strengths: I think the idea of explicitly modeling the missingness process is an important one which this work makes use of to good effect. The neural architecture here is designed to make use of fine-grained knowledge of video semantics and consequently, the model compares well against several baselines with good experimental results (particularly, those in Figure 3/4/5).

Weaknesses: I have a lot of unanswered questions on how this model was trained as well as the kinds of hyperparameters the learning algorithm was sensitive to.

Correctness: This is a little difficult to ascertain. There is a gap in this manuscript regarding details on how the model is trained. Line 153 says that the model is maximizing a variational lower bound and that details on the same were in the supplement but I could not find it. This is the sort of detail that should be in the main paper. Is there a prior distribution? If so, what is it? What kind of distribution is the one in Equation (5)? Does it represent a variational distribution? I am assuming the use of "q" is indicative of a variational approximation but this is not evident from what is written in the paper. This detail is relevant to understanding how the KL divergence (or entropy) term that is typically present in the variational bound is evaluated.

Clarity: * DIVE is presented as a deep generative model. However, I found the translation of the writing in Section 3.1-3.3 into a generative process quite difficult. * It is not clear how to backpropagate through FC(h_{iy}) in equation (4) since the heaviside step function is not differentiable. Did this pose and issue and if so, how was it handled? * How is p set in practice (equation (2))? * Line 140 talks about covariate shift but this is never elaborated on, where does the covariate shift come from? * In the synthetic examples and datasets, it is possible to know "N" (the number of objects) in a video. What happens when you do not know this number apriori?

Relation to Prior Work: Yes, there is a discussion of prior work.

Reproducibility: No

Additional Feedback: ** Post Discussion: Please incorporate all the writing changes and the additional experiments to the manuscript.


Review 3

Summary and Contributions: The paper introduces a novel method for model to learn video representation that disentangles pose, missingness, and appearance. The novelty lies in this missingness latent variable that is used to potentially impute the pose and appearance variable. Dynamic appearance is also introduce

Strengths: The method is mathematicaly sound and the experimental results seems to show that the proposed approach outperforms previous works.

Weaknesses: The main weakness lies in the evaluation. There is no ablation study. The model is more complicated than DDPAE and, without an ablation study, it's hard to tell if the method is better because of the better disentanglement or just because the architecture is bigger. I would recommend at least the following experiments: -measure the impact of dynamic appearance vs static: with LSTM, with a short time window, using a constant appearance -the overall same model but without imputing as described in 3.1 -what is the impact of the mixture in eq 2? what happens with different values of gamma In fig 4 and 5, if we compare the results of DDPAE with the results reported in the DDPAE's paper, the images are much worse. Why?

Correctness: Everything seems correct.

Clarity: Overall, the paper is clear. The figures 3,4,5 are not very clean. We see black rectangles around the text.

Relation to Prior Work: The presentation of related works is quite clear.

Reproducibility: Yes

Additional Feedback: The heavyside step function is not diferentiable. I don't see how backpropagation is performed. *****Post rebuttal****** I thank the authors for their very clear answers. They answered all my points. I raise my rating.


Review 4

Summary and Contributions: The paper deals with video prediction (on 2-digit Moving MNIST and MOTS data) in scenarios with "missing" data - 1) occluded pixels, 2) missing digits, 3) missed frames and deformed objects. It builds on top of the model presented in DDPAE [14] by including additional latent variables to account for and compensate for the missing of data. It shows the effectiveness of its model compared to previous variants in the above scenarios of missing data.

Strengths: The paper presents an effective way of tackling missing information, by referencing previous works that have done the same but perhaps not in the context of video prediction. The overall model is soundly designed to handle missing information, by both reconstructing the missing information in the input video as well as predicting for future frames. It is justified that the handling of missing data happens in the latent space rather than pixel space, and for the case of video most of the proposed ideas (distributions of latent variables, their connections, etc.) seem appropriate. The explanations of the method are quite clear and easy enough to understand.

Weaknesses: The model proposed was built on top of a model (DDPAE [14]) that was designed for and tested on simplistic datasets of Moving MNIST and Bouncing Balls. Hence, it is very effective on the simpler case of well-defined individual components in a dark background. It is encouraging that the model was able to achieve good results on this setting, it is to be seen how well it can perform in more complex datasets, such as those with natural images. The paper presents some motivating results on the MOTS dataset to address this very concern, however the method has the potential of working on more complex scenarios. In 3.2 Missingness Inference, the use of a normal distribution for sampling before the use of the heaviside step function is not quite justified.

Correctness: The claims of the paper are quite justified so far as the experiments designed are concerned. Three scenarios are proposed to check the missing value imputation the paper has introduced, and in all three the paper's method seems to work better than those methods that have not actively compensated for missing information.

Clarity: The paper is very well written, the explanation of the method is quite clear, the figures are very helpful.

Relation to Prior Work: Yes, it is clearly discussed how this paper is different from previous works, and it is mentioned that this paper builds on top of DDPAE [14].

Reproducibility: Yes

Additional Feedback: ------------------------ Here are my impressions from the author feedback: 1) I agree with R1 that DrNet should not have been mentioned as a stochastic method. Perhaps it is best to mention DrNet as a good deterministic model to compare with (notwithstanding the hierarchical deterministic methods that came after it). The authors have addressed this in their feedback. 2) R1:“we enforce separate representations for each object.” In lines 80-81 it is mentioned that “we assume that each video has a maximum number of N objects”. In the code, the option `n_components` seems to describe this max number of objects, which is used while initializing all the model priors in models/DiveModel.py. I agree with R2 that this is a convenient option though, so an ablation study on the number of objects that can be handled is useful. However, it is encouraging to see that the redundant objects are learned to be empty. 3) I agree with the authors that DDPAE can be considered the “exact model without missingness”. They provided the required experiment in the feedback. 4) The authors have clarified the issue with differentiating through the Heavyside function, as well as the nature of the prior distributions. Since they followed DDPAE, I assumed that they followed the same prior distributions as in DDPAE. 5) It is important to note the values of `p`, which the authors have clarified in their feedback. It should be included in the final draft. 6) I believe the authors have sufficiently answered all of R3’s clarification points in their feedback. 7) I agree with the other reviewers that the method has the potential of being effectively used in more complex datasets, such as at least KTH. However, this does not take away the introduced novelty or effectiveness of the method. I believe it has been sufficiently captured for the purpose of this submission. For now, I will keep my (top) rating as is.

[Author Response · NeurIPS 2020]

We thank the reviewers for their insightful comments. We are encouraged that reviewers found our paper clear, very well written, well motivated (**R1,R3,R4**). In particular, **R2** thinks the idea of explicitly modeling missingness is important. **R3** finds the missingness latent variable and dynamic appearance to be novel. **R4** recognizes the effectiveness of our method in reconstructing the missing information in the input video as well as predicting for future frames. Below we address specific concerns, and we will incorporate all feedback in the final version.

**R1, R3** *... exact same model architecture \*without\* the missingness* We consider DDPAE to be the closest architecture *without* the missingness variable. We also test the exact same model architecture with and without missingness variable for Scenario 2 of Moving MNIST, as shown in the table:

| Mov. MNIST Scenario 2 | BCE ↓ | | MSE ↓ | | PSNR ↑ | | SSIM ↑ | |
|---|---|---|---|---|---|---|---|---|
| Model (trained 300 epochs) | Rec | Pred | Rec | Pred | Rec | Pred | Rec | Pred |
| **DIVE w/o missingness** | 236.35 | 356.82 | 49.07 | 76.52 | 19.40 | 17.66 | 0.86 | 0.82 |
| **DIVE w missingness** | **165.42** | **321.29** | **27.03** | **64.17** | **22.15** | **18.56** | **0.93** | **0.83** |

**R2** *Line 153 says that the model is trained maximizing a variational lower bound...* Our model strictly follows the VAE framework (see caption of Fig. 1) by maximizing the ELBO objective as below:

$$
\begin{aligned}
\log p_\theta(\mathbf{y}^{1:K}, \mathbf{x}^{K+1:T}) \geq\ & \mathbb{E}_q \left[ \log p_\theta \left( \mathbf{y}^{1:K} | \mathbf{z}_{1:N}^{1:K} \right) - KL \left( q_\phi \left( \mathbf{z}_{1:N}^{1:K} \right) || p \left( \mathbf{z}_{1:N}^{1:K} \right) \right) \right] \\
+\ & \mathbb{E}_q \left[ \log p_\theta \left( \mathbf{x}^{K+1:T} | \mathbf{z}_{1:N}^{K+1:T} \right) - KL \left( q_\phi \left( \mathbf{z}_{1:N}^{K+1:T} \right) || p \left( \mathbf{z}_{1:N}^{K+1:T} \right) \right) \right]
\end{aligned}
$$

We will include these details in the main paper. Each component of $\mathbf{z}_i^t = [\mathbf{z}_{i,a}^t, \mathbf{z}_{i,p}^t, \mathbf{z}_{i,m}^t]$ is modeled separately.

**R2** *...details on how the model is trained.* Due to space limits, we deferred the training and datasets details to the supplemental material (supp.). We also included source codes to reproduce our results.

**R2, R3** *... the Heavyside function is not differentiable.* The Heavyside function $H(x)$ is **not** a variable node in our computational graph, hence we are NOT differentiating through it. We use *torch.where()* function in Pytorch to implement this condition operation. As shown in Fig. 1 top and Eq. (10), $\mathbf{z}_{i,m}$ is a masking indicator. In practice, we multiply each decoded object by the **logit** before the Heavyside function instead of the binary label (see Supp. L406 and L407). Hence, $\mathbf{z}_{i,m}$ gets its gradients from the decoder. We will clarify this in the updated version.

**R1, R2** *...knowing the number of objects .* The number of objects $N$ is specified a priori, but only as an upper bound. For the MOTS dataset, we set $N = 3$ but the actual number of objects is often lower. Similar to DDPAE, our model can learn to set the redundant components to be empty.

**R2** *What kind of distribution is the one in Equation (5)...Is there a prior distribution?* As stated in L121, Eq. (5) is the variational distribution. We use Gaussian as prior distributions, parametrized by mean and variance (L125, L138). The specific values for these parameters are provided in the code.

**R3** *there is no ablation study ... measure the impact of dynamic appearance vs static.* We **do** have ablation study in the Supp. due to space constraints. We **did** compare the impact of dynamic appearance modeling vs static in Supp., exactly as suggested by the reviewer.

**R2, R3** *How is the mixture p set in practice (eq (2))?...* For Eq. (2) we set $p = 0.25$ and for Eq. (8), $p = 0.85$ (see supp. L402-L404). We experimented with different values of p ($\pm 0.1$) and did not find any significant difference.

**R2** *talks about covariate shift but this is never elaborated on ...* The covariate shift comes from the distributional difference between training and testing data. Without the mixing strategy in Eq. (8), the model would overfit to the dynamic component of the appearance during training. Our random mixing regularizes the model to learn a better representation for static appearance.

**R3** *... In fig 4 and 5 ...results reported in the DDPAE's paper, images are much worse, Why?* Because our tasks are significantly more **difficult** than the experiments of DDPAE's paper. Fig. 4 and 5 show objects out-of-scene and dynamic appearance. DDPAE assumes that there is no missing object and that the appearance of each object remains static. Missing objects hinder the tracking and appearance modeling for DDPAE. If an object's appearance varies over time, DDPAE will learn an average appearance, leading to blurry reconstructions and errors in tracking.

**R1** *DRNet is not a stochastic video prediction method.* This is a typo. DRNet is a strong unsupervised video prediction baseline as it also learns disentangled representations from video.

**R4** *to be seen how well it can perform in more complex datasets* We appreciate your recognition of our contributions! We will consider more complex datasets and scenarios as future work, as also suggested by **R1**.

[Meta-Review · NeurIPS 2020]

After the authors response, all four reviewers have reached consensus and give the paper acceptance scores. All reviewers and the AC agreed that the authors’ response was very clear and addressed all raised questions. In particular the ablation of their model without missingness. Please update the paper with these new experiments as well as the changes mentioned in the rebuttal.